# Experimental Models to Define the Genetic Predisposition to Liver Cancer

**DOI:** 10.3390/cancers11101450

**Published:** 2019-09-27

**Authors:** Rosa M. Pascale, Maria M. Simile, Graziella Peitta, Maria A. Seddaiu, Francesco Feo, Diego F. Calvisi

**Affiliations:** Department of Medical, Surgical and Experimental Sciences, Via P. Manzella 4, 07100 Sassari, Italy; simile@uniss.it (M.M.S.); graziella.85@live.it (G.P.); maseddaiu@uniss.it (M.A.S.); feo@uniss.it (F.F.); calvisid@uniss.it (D.F.C.)

**Keywords:** hepatocarcinogenesis, genetic predisposition, quantitative trait loci, modifier genes, signal transduction, prognostic markers, interspecies comparison

## Abstract

Hepatocellular carcinoma (HCC) is a frequent human cancer and the most frequent liver tumor. The study of genetic mechanisms of the inherited predisposition to HCC, implicating gene–gene and gene–environment interaction, led to the discovery of multiple gene loci regulating the growth and multiplicity of liver preneoplastic and neoplastic lesions, thus uncovering the action of multiple genes and epistatic interactions in the regulation of the individual susceptibility to HCC. The comparative evaluation of the molecular pathways involved in HCC development in mouse and rat strains differently predisposed to HCC indicates that the genes responsible for HCC susceptibility control the amplification and/or overexpression of c-*Myc*, the expression of cell cycle regulatory genes, and the activity of Ras/Erk, AKT/mTOR, and of the pro-apoptotic Rassf1A/Nore1A and Dab2IP/Ask1 pathways, the methionine cycle, and DNA repair pathways in mice and rats. Comparative functional genetic studies, in rats and mice differently susceptible to HCC, showed that preneoplastic and neoplastic lesions of resistant mouse and rat strains cluster with human HCC with better prognosis, while the lesions of susceptible mouse and rats cluster with HCC with poorer prognosis, confirming the validity of the studies on the influence of the genetic predisposition to hepatocarinogenesis on HCC prognosis in mouse and rat models. Recently, the hydrodynamic gene transfection in mice provided new opportunities for the recognition of genes implicated in the molecular mechanisms involved in HCC pathogenesis and prognosis. This method appears to be highly promising to further study the genetic background of the predisposition to this cancer.

## 1. Introduction

Hepatocellular carcinoma (HCC) is a frequent human cancer, with 0.25–1 million new cases being diagnosed each year [1,2,3]. The highest frequencies of HCC occur in Sub-Saharan Africa and far eastern Asia, due to endemic infections by hepatitis B virus (HBV) and hepatitis C virus (HCV), and in regions where food is contaminated by Aflatoxin B1, a potent genotoxic mycotoxin produced by *Aspergillus Flavus* [1,2,3,4,5]. Furthermore, HCC incidence is increasing steadily, even in countries where the occurrence is relatively low [1,6,7,8]. In addition, numerous cohort and case-control studies demonstrated that liver cirrhosis caused by alcohol consumption and cigarette smoking significantly augments the risk of liver cancer [7,8,9].

Mounting evidence indicates that numerous monogenic and polygenic diseases are at play in the predisposition to HCC [10]. They include: genetic diseases such as autoimmune hepatitis, type 2 diabetes, a family history of HCC, hereditary thysosinemia acuta, porphyria acuta intermittens, and porphyria cutanea tarda, α1-antitrypsin deficiency, different types of glycogen storage disease, hemochromatosis, autoimmune hepatitis, and the metabolic syndrome. However, the existence has been documented of an individual genetic predisposition to HCC that raises the risk of this tumor, even in individuals not subjected to known predisposing factors [5].

This review explores the genetic mechanisms that control the inherited predisposition to HCC implicating gene–gene and gene–environment interactions.

## 2. The Individual Susceptibility to Liver Cancer

The existence of an inter-individual susceptibility to the development of HCC in rodents and humans is well-known [11], and at least in part, it depends on inter-individual variations in the capacity to activate the carcinogens [12,13,14,15,16,17]. However, numerous studies on experimental hepatocarcinogenesis have revealed that the different susceptibility of various mouse and rat strains to liver cancer may depend on changes in the capacity of initiated cells to evolve to preneoplastic and neoplastic lesions [11,18,19,20,21,22,23]. The number of initiated cells in urethane-treated resistant mice strains, C57BL/6JxBALB/c (B6C) and C57BL/6J x Mus spretus (B6S), does not differ consistently from that of the parental susceptible strains, but the preneoplastic lesions induced in these mice have a low capacity to evolve to HCC [18,19,20,21,22]. The ODS and NAR rat strains, and some Wistar strains, resistant to hepatocarcinogenesis, exhibit a higher inducibility of cytochrome P-450 isoforms, necessary for carcinogen activation, than the susceptible strain Fisher 344 (F344) [23]. Also, the resistant DRH rat strain was established by inbreeding a closed colony of Donryu rats for >20 generations continuously feeding 3V-methyl-4-dimethylaminoazobenzene (3V-Me-DAB) and selecting for reduced HCC incidence during inbreeding for >10 years [24]. In these rats, treated with 3-methylcholanthrene to induce carcinogen metabolizing enzymes, no significant increase in the in vivo formation of DNA adducts with 3V-Me-DA was detected [25]. Although no significant differences in the hepatocarcinogenesis initiation by diethylnitrosamine (DENA) and 3V-Me-DAB were observed between DRH rats and susceptible F344 rats [26], the FAH induced in DRH rats were found to be less prone to progress to HCC than the F344 rats counterparts [26,27]. The Brown Norway (BN) [28] and the Copenhagen (Cop) [29,30] rat strains, well-known to be strongly resistant to hepatocellular carcinogenesis, after crossing with F344 rats, dominantly transmit their resistance to (BNxF344)F1 (BFF1) and (CopxF344)F1 (CFF1) progeny. When these rats are subjected to DENA/AAF/partial hepatectomy, according to the ‘‘resistant hepatocyte’’ protocol of hepatocarcinogenesis [31], an elevated number of fast-growing early preneoplastic liver lesions that, however, after exhaustion of the promoting stimulus, exhibit a progressive decrease in growth capacity and phenotypic reversion (a phenomenon known as “remodeling”).

## 3. The Genetic Model to Study the Rodent Susceptibility to Liver Carcinogenesis

Studies aimed at revealing the genetic mechanisms responsible for differences in phenotypic susceptibility to liver carcinogenesis hypothesized that various “modifier” genes alter the expression of other genes, thus affecting different stages of tumorigenesis and the severity of cancer [32]. The strategy to identify modifier genes implicates the analysis of mouse or rat strains with different susceptibilities to HCC in order to define the inherited predisposition, at molecular and cellular levels, and generate backcrosses or intercrosses, in which allelic variants segregate at each locus. The genotypic analyses of these populations were conducted to identify quantitative trait loci (QTLs) and genetic interactions interfering with cancer predisposition. Putative candidate modifier genes could be identified according to their position in QTLs and their functional activity. However, the linkage-analysis to identify QTLs allows mapping loci of 10–30 cm, much larger than the 1–2 cm intervals necessary for cloning tumor modifier genes. The restriction of QTLs area and the analysis of haplotype, linkage disequilibrium studies, and gene expression profiling could favor the identification of modifier genes.

Recombinant and ‘‘congenic’’ and “consomic” strains have been generated to this purpose. A strategy to restrict the QTL area is the generation of recombinant congenic strains (RCSs) [33] by a program allowing the casual segregation of 12.5% of loci from a donor strain on another isogenomic milieu. This allows the generation of a subgroup of RCSs in which the polygenic trait becomes oligogenic. In each RCS, present homozygous susceptibility or resistance alleles are present, derived from the donor strain at a given locus, whereas the genetic background of the other strain is present in the remaining genome (Figure 1). Therefore, RCSs may be constructed with a genetic background that is prevalently either susceptible or resistant. The subsequent generation of sub-congenic strains, with smaller portions of QTL introgressed from the donor congenic strain onto the isogenomic background, increases the possibility of the positional cloning of QTL genes. A less time-consuming and expensive approach is the construction of consomic strains (CSs), where an entire chromosome is introgressed into the isogenomic background of another inbred strain [34,35] (Figure 2). Chromosome substitution is attained by numerous backcrossing of individuals heterozygous for the selected chromosome to the recipient strain. Thus, RCSs are generated from CSs over a restricted section of a given chromosome, and QTLs are located, in animals with a fixed background, by F2 linkage analysis [36]. QTL mapping in consomic animals permits the identification of QTLs with small phenotypic effects. On the other hand, however, it does not offer any advantage for gene cloning. A combination of linkage analysis and linkage disequilibrium [37] has allowed the reduction of QTLs intervals to 1–2 cm using outbred Mus spretus mice in crosses with inbred Mus musculus [38].

Recombinant inbred strains, descendants from two inbred strains, are mosaics of the founder haplotypes that allow establishing the strain distribution patterns of each marker and its correlation with the phenotype in each recombinant strain. Linkage disequilibrium is defined as the excess of co-occurrence of two alleles over the expectance if the two alleles were independent. Via linkage disequilibrium studies, mapping of the mutations responsible for a given phenotypic trait is possible: in fact, the markers closest to the gene involved show the strongest correlation with the phenotype.

## 4. The Loci Controlling the Susceptibility to HCC

The first approach to clarify the mechanisms of the susceptibility to liver cancer was the recognition of the QTLs in crosses of phylogenetically distant mice and rats. In urethane-treated F2 male mice, generated by crossing the susceptible C3H/HeJ strain with the resistant A/J strain, the hepatocarcinogenesis susceptibility loci Hcs1, Hcs2, and Hcs3, positioned in chromosomes 7, 8, and 2, respectively, were identified [22] (Table 1). Intercrosses between C3H/HeJ and Mus spretus, phylogenetically distant mice, followed by the cross of the resulting F1 with the resistant C57BL/6J (B6) strain, resulted in the identification of additional loci, namely Hcs4, Hcd5, and Hc6 loci, located in chromosomes 2, 5, and 19, respectively (Table 1). The analysis of the backcrosses and intercrosses between the susceptible C3H/HeJ or CBA/J strains and the resistant B6 strain led to the discovery of Hcs7, mapping to distant chromosome 1 [22] (Table 1).

Congenic B6.C3H(D1Mit5-D1Mit17) and B6.BR(D1Mit5-D1Mit17) mice were generated, in which a 70 cm segment (between D1Mit5 and D1Mit17) from C3H or C57BR/cdj (BR) susceptible strains was introgressed onto a B6 background. The generated RCSs developed more hepatocellular tumors than B6 mice, implying that a distal portion of chromosome 1 carries modifier gene(s) conferring susceptibility to liver cancer. Two loci involved in the susceptibility to HCC were identified in crosses between BR and B6 mice [39]. BR females are exquisitely prone to HCC induction, since they are genetically insensitive to the inhibition of hepatocarcinogenesis exerted by ovarian hormones. This property was dominantly transmitted to B6BRF1 mice. BR alleles at two loci, on chromosomes 17 and 1 (Table 1), identified in backcrosses and F2 progeny, were associated with increased susceptibility in both sexes. They were denominated Hcf1 and Hcf2 (hepatocarcinogenesis in females) loci. Hcf1 and, at a lower extent, Hcf2, accounted for the higher susceptibility of BR mice.

In addition to susceptibility loci, two hepatocarcinogenesis resistance loci, Hcr1 and Hcr2 loci with negative phenotypic effects, mapping on chromosomes 4 and 10, respectively, were discovered in mouse genome [40] (Table 1). The resistance alleles were provided by the DBA/2J mice strain, susceptible to hepatocarcinogenesis during perinatal life, but resistant during adult life [41,42]. In these mice, susceptibility loci possessing very limited phenotypic effect were present [40], but resistant F1 mice were generated by crossing the resistant BXD-15 recombinant inbred mouse, carrying Hcr genes, wirh DBA/2J carrying Hcs genes [43]. This suggests that Hcr genes may modify the activity of sensitivity loci.

The studies on rat hepatocarcinogenesis have clearly shown that hepatocarcinogenesis is a multistep process: cells initiated by chemical carcinogens form, in rat liver, small aggregates of few cells, and minifoci of 10–100 cells positive to the immunohistochemistry by the placental isoform of glutathione-S-transferase (GST). The proliferation of these cells leads to the progressive formation of foci of altered hepatocytes (FAH), dysplastic nodules (DN), and HCC (Figure 3). During this process, some cells apparently disappear because of re-differentiation (remodeling) [44]. Remodeling nodules are identified as areas lacking uniformity of GST-P immunostaining and with irregular margins (Figure 3). Remodeling progressively decreases, whereas cell proliferation increases during the evolution of FAH to HCC. Previous work in our laboratory has shown that initiated cells evolve rapidly to HCC and remodeling is relatively exiguous in rats genetically susceptible to hepatocarcinogenesis, whereas in some resistant rat strains, the evolution of initiated cells proceeds slowly, many preneoplastic lesions remodel, and only few HCC are formed [11].

A locus (rcc), positioned on the telomeric end of chromosome 20, is present in different MHC-recombinant rat strains, congenic for the MHC (major histocompatibility) genes present in the grc (growth reproduction complex) region [45] (Table 2). Of note, this locus has many properties in common with tumor-suppressor genes: indeed, it is recessive, its deletion causes phenotypic susceptibility to various carcinogens, and inhibits tumor development in many organs and tissues, including liver, skin, kidney, and the mesenchyme [45].

Further research revealed that numerous loci regulate the polygenic predisposition to rat HCC. Linkage analysis experiments with male backcrosses and intercrosses of the resistant BN of Cop rats to the susceptible F344 rats (Table 2) allowed the identification, in BN × BFF1 backcross progeny, of the Hc1 and Hcs2 loci, respectively in chromosomes 7 ad 1 [46], Hcs3 and Hcs4 loci in BFF2 rats [47], and Hcs4, Hcs5, Hcs6, and Hcs7 in CFF2 intercrosses [48]. Furthermore, in BN × BFF1 backcrosses, the Hcr loci 1, 2, and 3 were mapped to chromosomes 10, 4, and 8, respectively [47]. Moreover, the Hcr loci, 9 to 12 (formerly known as Hcr 4–7), were mapped on chromosomes 4, 6, and 8 of BFF2 rats [47], while Hcr 13 and 14 (previously called 8 and 9) were localized on chromosomes 4 and 18 of CFF2 rats [48]. Furthermore, the Drh1 and Drh2 loci were discovered on chromosomes 1 and 4 of (DRHxF344)F2 rats [26,27]. These loci regulate the development of FAH induced by 3V-Me-DAB [26,27]. Based on the chromosomal localization, Drh1 corresponds to Hc3 and Hc5, while Dhr2 corresponds to Hcr2.

Neoplastic nodules induced in the hybrid BFF1 rats, which are produced by crossing the susceptible F344 and resistant BN strains, display low DNA synthesis and a high degree of remodeling. To identify the loci controlling the remodeling of nodules induced by the RH protocol 32 weeks after initiation with DENA, whole-genome scanning of BFF2 rats was performed (Table 2). Two loci in suggestive linkage with the percentage of remodeling nodules were identified on chromosomes 7 and 1 in BFF2 rats and defined loci Lnnr 1 and 2 (RGD; previous denomination, Hcrem1 and Hcrem2). These loci reduced the percentage of remodeling lesions in Cop rats [47]. In CFF2 rats, Lnnr3, on chromosome 2, reduced the number of remodeling DNs, whereas Lnnr 4 and 5, on chromosome 13, increased the number of these nodules. Positioned on chromosome 13 was also Lnnr6, which reduced the volume of remodeling nodules [49].

The Hcs4 locus of BFF2 male rats [47], spanning about 20 cm on the centromeric side of chromosome 16 with a LOD score peak at 9.04 cm, regulates the volume of neoplastic nodules. The determination of the average phenotypic value and allelic distribution pattern, in the homozygous and heterozygous progeny, showed nodules with higher volumes in the rats carrying 1/2 B alleles at this locus than FF homozygous rats [47]. A congenic rat line was constructed [50] by transferring the Hcs4 BN allele onto a F344 genetic background, and by marker assisted selection of crosses, the Hcs4 locus was narrowed to 4.41 cm, including the LOD score peak at 9.04 cm from the centromere. The male rats of this RCS, (F344.BN-Hcs4) were highly susceptible to hepatocarcinogenesis, whereas female rats were as resistant as female BN rats. The gonadectomy of recombinant rats greatly reduced the susceptibility of males and increased that of females indicating that the BN strain contributed hormone responsive resistance alleles, responsible for the resistance of female rats, the activity of which was inhibited by male sex hormones and increased by female sex hormones. These observations indicated the presence on chromosome 16 of one/more genes conferring resistance to hepatocarcinogenesis to female rats. This important discovery represents the first demonstration that sex hormones may regulate modifier genes controlling the predisposition to liver carcinogenesis.

The studies of the genetic predisposition in mouse and rat models to hepatocarcinogenesis confirmed the polygenic nature of the process, showing that these genes influence the promotion and progression stages of the process by modulating cell proliferation and cell remodeling/apoptosis. The phylogenetic tree of various rat strains [51] demonstrated that during the generation of the F344 strain, from a common resistant feral ancestor, the resistance alleles underwent numerous selective mutations that, consequently, could not be activated by carcinogens. Reciprocal epistatic interactions, influencing the number end volume of preneoplastic nodules, have been found between microsatellite marker loci. The QTLs identified in BFF2 and CFF2 rats have individually relatively poor phenotypic effects. However, the valuation of epistatic interactions between microsatellite loci, the phenotypic effects of which cannot be predicted by the sum of their separate effect, led to the identification of novel tumor modifier loci, involved in the determination of the number and volume of lesions [46,47,48], suggesting that the interactions between susceptibility genes have a role in hepatocarcinogenesis. This is consistent with the observation that in BN x BFF1 backcrosses larger neoplastic nodules occurred in heterozygous FB rats, than in homozygous BB rats, indicating that the maintenance of unaltered resistance alleles in BN rats inactivates the susceptibility alleles. The existence of many Hcs, Hcr, and Lnnr loci and of epistatic loci, implicated in the same causative pathways, reveals the high complexity of the genetic factors involved in hepatocarcinogenesis: the expression of a number of putative suppressor genes modifies the positive phenotypic effects of susceptibility alleles.

## 5. The Genes Involved

The progression of liver preneoplastic lesions is associated with considerable genetic instability [52]. The presence of multiple copies of chromosome 1 or a duplication of a region of this chromosome, along with rat hepatocarcinogenesis, suggests the presence of genes involved in the neoplastic transformation, and the loss of 3p and the last band of 6q suggests the presence of oncosuppressor genes [52].

Available data show that numerous QTLs are involved in susceptibility to HCC in mice and rats. However, susceptibility genes have not yet been identified. Available data suggest that some of these genes are plausible candidates. The genes present in QTLs include oncogenes such as *H-Ras*, *Myc*, *Jun*, *Fos*, *Raf1*, *Met*, *Odc*, *Akt2*, *Akt3*, *Itpr1*, *Jak2*, *Esr*, *Esr2*, oncosuppressor genes, such as *Tsg101*, *Pten*, *Cdkn1c*, *H19*, *Dcc*, growth factors and growth factor receptor genes, including *Igf1*, *Igf2*, *Igf2r*, *Tgf-a*, *Egfr*, and genes involved in cell death (*Bax*, *Tgfb1*, *Tnf-a*) and DNA repair (*Gadd45a*, *Adprt*) [11]. Some of these genes could be involved in the determination of a susceptible or resistant phenotype, but there is no proof that they are modifier genes responsible for the genetic susceptibility to HCC.

In an attempt to better characterize susceptibility/resistance genes, the hybrid LFF1 strain was generated by crossing the susceptible Long-Evans (LE) and F344 rats, and poorly, moderately, and well-differentiated HCCs were tested for allelic imbalance (AI) at chromosomes 1, 4, 7, 8, and 10, where Hcs/Hcr loci are located, and at chromosomes 3 and 6, where deletions have been found in cytogenetic studies [53]. Linkage analysis identified Hcs1 and 2 loci on rat chromosomes 7 and 1, and Hcr1, 2 and 3 loci, on chromosomes 10, 4, and 8, respectively. A study was performed on HCCs induced in F1 hybrid between susceptible Long-Evans (LE) and F344 rat strains in the attempt to discover possible modifier genes [53]. The hybrid rats showed allelic imbalance (AI) at multiple regions located on chromosomes 6, 7, and 10. Detailed deletion mapping of chromosome 10 localized a putative suppressor Hcr1 gene within a 3.2-cm interval and two other regions with frequent AI in ~40% of HCCs. AI also was detected at the p53 locus and at the Hcs1 locus on chromosome 7, where the protooncogene c-Myc is located. Of note, the latter gene is often amplified and overexpressed in HCC [54,55]. Interestingly, most AIs occurred in poorly/moderately differentiated HCCs.

The stearoyl-CoA desaturase 1 (Scd1) gene, implicated in the synthesis and regulation of unsaturated fatty acids, is upregulated in treatments associated with hepatocarcinogenesis, such as peroxisome proliferators, iron overload, and dichloroacetic acid [56]. We found that *Scd1* expression is significantly higher in the liver of C3H/He mice and F344 rats, genetically susceptible to hepatocarcinogenesis, than in the liver of resistant BALB/c mice and BN rats [56]. The Scd1 locus does not exert allele-specific effects in a BALB/cxC3H/He intercross and in a BNxF344 backcross and intercross. Furthermore, *Scd1* coding polymorphisms occur in the mouse and the rat strains exhibiting *Scd1* upregulation. These findings together exclude *Scd1* candidacy as an HCC-modifier gene, while they suggest that the *Scd1* gene is presumably a downstream effector of an unidentified modifier gene/locus in rodents.

Further attempts to identify modifier genes, based on the analysis of single nucleotide polymorphisms, were carried out [57]. The genome of the BALB/c mouse strain provides alleles that dominantly inhibit the development of HCC in F1 crosses with the susceptible C3H/He strain. Genome-wide linkage analysis using a 1536-single-nucleotide polymorphism array in a (C3H/Hex BALB/c)F2 intercrosses, treated with urethane to induce HCC, identified the Hpcr3 resistance locus, which maps to the central part of chromosome 15 and accounts for about 40% of the phenothypical variance [57]. Importantly, this locus is situated in a region homologous to rat Hcs1, which influences the promotion and progression stages of liver carcinogenesis. The BALB/c-derived allele at the Hpcr3 locus reduced tumor-occupied area up to 25-fold, in a semi-dominant way. A transcriptomic analysis of normal mouse liver revealed the association of Hpcr3 with the susceptibility of BALB/c, C3H/He, and F1 mice to hepatocarcinogenesis and identified the genes present in the Hpcr3 locus. According to this analysis, the E2F1 pathway was found to be implicated in the modulation of the susceptibility to hepatocarcinogenesis [57].

## 6. Phenotypic Effects of the Variation of the Genetic Susceptibility to Liver Cancer

The genes that determine the genetic susceptibility to HCC influence the expression of key regulatory genes of different signal transduction pathways implicated in hepatocarcinogenesis. The study of preneoplastic and neoplastic lesions induced in rat strains differently susceptible to hepatocarcinogenesis allowed the discovery of the effects of cancer modifier genes on these pathways.

### 6.1. The Role of c-Myc

The highest amplification and/or overexpression of c-*Myc*, a gene positioned in Hcs1, was observed in DNs and HCCs, chemically induced in F344 rats, genetically susceptible to hepatocarcinogenesis [11], compared to slowly progressing lesions developed in BN and Wistar resistant strains [55]. Interestingly, c-Myc amplification is implicated in the malignant conversion in human hepatocarcinogenesis as well [58]. After the inactivation of the c-Myc transgene, there occurs the tumor cells re-differentiation in c-Myc transgenic mice [59]. Furthermore, according to recent observations, the inhibition of heat shock factor 1 (HSF1), a positive regulator of mTORC1, reduces c-Myc expression and restrains the proliferation of c-Myc-derived mouse HCC cell lines [60]. *In vivo*, the hydrodynamic delivery of a dominant negative form of HSF1 (HSF1dn) in the mouse inhibits the hepatocarcinogenesis induced by c-Myc hyperexpression. [60]. Studies in c-Myc-driven mouse HCC [61] showed the activation of TORC2 with consequent phosphorylation/activation of Akt1, but not Akt2. The loss of Akt1, but not that of Akt2, prevented c-Myc-driven mice hepatocarcinogenesis. Silencing of Rictor or Akt1 in c-Myc overexpressing HCC cell lines inhibited p-Foxo1 expression and suppressed cell growth. In c-Myc mice, the inhibition of mTORC1 prevented c-Myc-driven HCC progression, whereas the inhibition of both mTORC1 and mTORC2 by MLN0128 induced apoptosis and necrosis of tumor tissue.

### 6.2. The Ras-Family Genes and Related Pathways

*Ras*, another gene positioned in Hcs1, is upregulated in the DN and HCC of F344 rats. The DN and HCC of genetically susceptible F344 rats exhibit deregulated G1 and S responsible for their low progression [62]. A study on the implication of inhibitors of Ras/Erk signaling in the acquisition of resistance or susceptibility to hepatocarcinogenesis showed that a moderate activation of Ras, Raf-1, and Mek proteins is associated, in both F344 and BN rats, to a robust induction of Dab2 and Rkip inhibitors [62]. The levels of Dusp1 increased only in BN rat lesions, in which modest ERK activation occurred, while a strong Dusp1 decline was found in the corresponding lesions from F344 rats, in which an elevated ERK activation occurred [62]. Furthermore, both F344 and BN rat strains showed a gradual increase of apoptosis driven by RassF1A/Nore1A/Mst1 (Ras association domain family 1A/novel Ras effector 1A/mammalian sterile twenty kinase 1) in DNs and HCCs, with the highest levels of gene expression and apoptosis in BN rat HCC, whereas loss of Dab2IP (Dab2-interacting protein), a protein implicated in Ask1 (apoptosis signal regulating kinase 1)-dependent cell death, was only found in F344 rat malignant lesions [62] (Figure 4). This situation indicates that HCC susceptibility genes control the Ras/Erk axis and the pro-apoptotic Rassf1A/Nore1A and Dab2IP/Ask1 pathways. Dusp1 has a prominent role in the acquisition of the resistance of BN rats to HCC development, while the late activation of RassF1A/Nore1A and Dab2IP/Ask1 pathways are responsible for the higher apoptosis observed in BN HCC [62].

Activation of the Ras/Erk pathway leads to the induction of a plethora of downstream effectors, including *Foxm1* (forkhead box M1, Figure 5), in various experimental and human tumor types. Once activated, *Foxm1* induces the transcription of *Aurka* (Aurora A) and *Nek2* (never in mitosis gene A-related kinase 2) genes, involved in genomic instability, as well as of *Cyclin B1*, *Cdc2*, and *Cdc25b* (cell division cycle 25B) that regulate G2-M transition, and of the antiapoptotic *Survivin* and angiogenesis genes such as *Erytropoietin* and *Vegf* [63]. Furthermore, Foxm1 activates the Skp2-Csk1 ubiquitin ligase, thus determining the proteasome-dependent degradation of the Erk1/2 inhibitor Dusp1 [64] (Figure 5). Foxm1 and its targets are up-regulated earlier and at higher level in DNs and HCCs of the susceptible F344 than the resistant BN rats, thus contributing to the higher aggressiveness of F344 rats, compared to BN rats [64].

Different observations indicate that *SKP2* is overexpressed in experimental and human HCCs. A positive correlation between the nuclear SKP2 positivity and the clinical aggressiveness of HCC and shorter patient survival has been found [65]. Accordingly, we demonstrated the existence in HCC of a genetic control of the degradation by the SKP2/CSK1 ubiquitin ligase of the cell cycle regulating proteins p21^WAF1^, p27^KIP1^, p57^KIP2^, and p130 that contributes to determine the susceptibility to hepatocarcinogenesis [66] and HCC prognosis [67]. Responsible for *SKP2* overexpression in HCC could be the disruption of the negative control operated by KIF14 (kinesin family member 14) [68]. Also, it has been found that core promoter mutations of HBV contribute to HCC development by the SKP2-dependent degradation of the *p21 ^j^* oncosuppressor gene [69].

To further define the SKP2 role in hepatocarcinogenesis, *SKP2* stable overexpression was induced, through hydrodynamic gene delivery, in the mouse liver, either alone or in combination with activated forms of N-*Ras* (N-RasV12), *Akt1* (myr-Akt1), or *β-catenin* (ΔN90-β-catenin). It was observed that the forced overexpression of *Skp2*, *N-RasV12*, or Δ*N90-β-catenin*, alone or the co-expression of *Skp2* and Δ*N90-β-catenin* did not induce liver tumor development, while overexpression of *myr-Akt1* alone induced HCC development after long latency [70]. In contrast, the co-expression of *SKP2* and N-*RasV12* or *myr-AKT1* induced the early development of multiple HCCs in all SKP2/N-RasV12 and SKP2/myr-AKT1 mice [70]. At the molecular level, a strong induction of AKT/mTOR and Ras/MAPK pathways occurred in preneoplastic and neoplastic liver lesions from SKP2/N-RasV12 and SKP2/myr-AKT1 mice. In addition, although the oncogenic power of *Skp2* seems to depend on its ability to induce the degradation of the tumor suppressor proteins p27, p57, Dusp1, and Rassf1A, the same proteins did not decline in liver lesions from SKP2/N-RasV12 and SKP2/myr-AKT1 mice, suggesting a suppressor activity independent of their degradation [70]. Accordingly, SKP2 deficiency inhibits skin carcinogenesis in a p27-independent way [70]. Thus, on the basis of these observations, it may be concluded that a cooperation of SKP2 with N-Ras and AKT oncogenes promotes liver cancer in mice, in agreement with the observation that the nuclear translocation of SKP2 is associated, in human HCC, with activation of the AKT/mTOR and Ras/MAPK pathways.

### 6.3. Mybl2 and AKT and ERK1/2 Signaling

Cyclin D1 overexpression, associated with overexpression of MAP kinases (ERK1/2, p38 and JNK1/2) Akt (v-akt murine thymoma viral oncogene homolog 1), Pak1 (p21-actived kinase19), and inactive Gsk3β (glycogen synthase kinase-3β), occur in chemically-induced preneoplastic and neoplastic lesions of rat liver [71]. Further, activation of the Akt/PKB (protein kinase B) occurs in HCC growing in c-Myc/TGF-α double transgenic mice [72]. In mouse HCC induced by diethylnitrosamine, low expression of the potent ROS scavenger Metallothionein depends on the negative regulation by PI3K/AKT signaling pathway [73]. Furthermore, AKT and N-Ras (neuroblastoma ras viral oncogene homolog) coactivation induces mouse hepatocarcinogenesis through the mTOR (mammalian target of rapamycin complex 1), FOXM1/SKP2, and c-Myc pathways [74]. HCC development is also promoted by the co-activation of AKT and c-Met through the activation of the c-Myc and the mTORC1/FASN and FOXM1/SKP2 pathways [75]. Accordingly, in a mouse model, generated by hydrodynamic gene transfer, leading to the overexpression of both activated AKT and neuroblastoma Ras viral oncogene homolog (N-Ras) in the liver, AKT and N-Ras expression accelerated much more HCC development by activating the mammalian target of rapamycin complex 1 (mTORC1) than the overexpression of AKT alone [75].

A connection between the expression of the transcription factor MYBL2 and AKT and ERK1/2 signaling has been suggested. Higher MYBL2 expression was found prevalently in the nuclei of DN and HCC of F344 rats, and in HCC of E2F1 transgenic mice, than in slow progressing corresponding lesions of BN rats and c-Myc transgenic mice [76]. Furthermore, in fast progressing DN and HCC of E2f1 transgenic mice, MYBL2, Clusterin, Cdc2, and Cyclin B1 expression was higher than in the lesions of c-Myc transgenic mice, and anti- MYBL2 siRNA had the highest anti-proliferative and apoptogenic effects in cell lines from HCC of E2f1 transgenic mice. MYBL2 transfection in HepG2 and Huh7 liver cancer cells enhanced the proliferation and G1/S and G2/M cell cycle phase transition, while the opposite occurred when MYBL2 expression was inhibited by specific siRNA [77]. MYBL2 transfection in Huh7 liver cancer cells activated genes involved in cell proliferation, such as *MDK* (Midkine) [77], an activator of AKT and ERK1/2 pathways [78] (Figure 6). Gene expression profiles, comparatively done in MYBL2-transfected Huh7 cells, displayed the upregulation of signal transduction and cell proliferation genes (*MYBL2*, *GIPR*, *RHO*, *RPS27*, *CSNK1D*, *ODC1*, *NUDC*, and *MDK*), upregulation of the transcription regulator HDAC10, and cell motility (*TPM4*, *TUBA1C*), and downregulation of the oncosuppressors *PPP1CA*, *MRPL41*, and *HINT1* genes [77]. Notably, the Hdac10 protein expression was found to progressively increase, whereas Pp1CA expression progressively decreased from normal liver to DN and HCC of c-Myc and *E2f1* transgenic mice, and the highest changes were found in the more aggressive HCC of *E2f1* mice [79]. Pp1CA, by inhibiting AKT phosphorylation at Thr-450, restricts the capacity of the PI3K/AKT signaling to promote cell survival and proliferation by stimulating WNT/β-catenin and IKK/NF-kB pathways [80].

Interestingly, a mouse model of hepatocarcinogenesis in which the combined overexpression of activated mutant forms of *Pik3ca* (PIK3CAH1047R) and *Yap* (YapS127A) were induced by hydrodynamic transfection (Pik3ca/Yap) [81] showed that the oncogenic cooperation of *Pi3k* and *Yap* led to the activation of the mTORC1/2, ERK/MAPK, and Notch signaling. The simultaneous activation of PI3K and Yap pathways is frequent in human HCC and their joint inhibition strongly suppresses the *in vitro* growth of HCC and CCA cell lines [81].

### 6.4. The Methionine Cycle

The methionine cycle plays a fundamental role for cell growth and defense against peroxidative liver damage. In this cycle, methionine is converted to SAM (S-adenosylmethionine) by methionine adenosyltransferases: MATI/III and MATII. SAM is used in methylation reactions, catalyzed by various methyltransferases or GNMT (glycine methyltransferase), and transformed to SAH (S-adenosylhomocysteine) (Figure 7). SAHH (S-adenosylhomocysteine hydroxylase) transforms SAH to homocysteine. The latter may be transformed to cystathionine by a β-synthase, followed by the synthesis of reduced glutathione, or for methionine resynthesis. The latter may occur during the synthesis of phosphatidylethanolamine from phosphatidylcholine, catalyzed by PEMT (phosphatidylethanolamine methyltransferase), in the Bremer pathway [82]. The transformation of phosphatidylcholine to choline followed by its conversion to betaine is coupled to the transformation on homocysteine to methionine in a reaction catalyzed by betaine homocysteine methyltransferase. Alternatively, the transformation of homocysteine to methionine is coupled to the folate cycle, in which THF (tetrahydrofolate) is transformed to CH_2_-THF (5,10-methenyltetrahydrofolate), in a reaction catalyzed by methyltetrahydrofolate reductase, coupled with the resynthesis of glycine from sarcosine. It follows the synthesis of CH_3_-THF (5-methyltetrahydrofolate), catalyzed by 5,10-methylene-tetrahydrofolate reductase, and the conversion of CH_3_-THF to methionine by methionine synthetase (Figure 7). Finally, SAM, decarboxylated by a specific decarboxylase, is used for polyamine synthesis.

Low SAM levels favor homocysteine re-methylation, and then glutathione synthesis, while high SAM levels activate cystathionine β-synthase, whose Km for SAM (1.2–2 mm) is higher than that of methyltetrahydrofolate reductase (60 μm) [83]. SAH is a potent competitive inhibitor of transmethylation reactions, also inhibited by 5’-methylthioadenosine (MTA), a reaction product of polyamine synthesis.

In mammals, the liver-specific *MAT1A* gene encodes the MATI/III isoforms, and the widely expressed *MAT2A* gene encodes MATII isozyme [84]. MATI and MATIII isozymes have 23 μm–1 mm and 215 μm–7 mm Km for methionine, respectively, while the MATII isoform has the lowest Km for methionine (4–10 μm) [83]. The SAM physiologic level (~60 μM) inhibits the *MAT1A* isoform and favors MAT2A activity. 

Chronic hepatitis and cirrhosis and HCC of rodents and humans are characterized by a fall in *MAT1A* expression and a rise in *MAT2A* expression, with consequent decrease of *MAT1A*: MAT2A ratio (an event referred to as “*MAT1A*/MAT2A switch”) [83,85]. MATII up-regulation cannot compensate for the decrease in MATI/III isozyme because of inhibition of MATII isozyme by the reaction product [85]. The fall of MATI/III: MATII activity ratio robustly contributes, together with the rise in SAM decarboxylation for polyamine synthesis, to the strong decrease in SAM levels [86,87].

The injection of SAM to rats after the end of treatments with carcinogens prevents HCC development [84,86,88,89,90,91,92]. Accordingly, the transfection of *MAT1A* in human HCC cell lines or the addition of SAM to the culture medium strongly inhibits cell proliferation. [93,94]. Also, tumor development in rat liver parenchyma, after the injection of the human HCC cell line H4IIE, is inhibited by continuous SAM intravenous infusion [92]. It should be noted that SAM administration to these rats is not curative due to the compensatory induction of hepatic GNMT expression that prevents SAM accumulation [92]. It should be interesting to assess the effect of SAM administration to HCC patients in which GNMT expression is silenced.

These important findings strongly suggest the involvement of the *MAT1A*/MAT2A switch and SAM fall in hepatocarcinogenesis. This was definitively and more directly demonstrated by the observation that MAT1A knockout mice, characterized by chronic SAM deficiency, exhibit hepatomegaly at 3 months of age, extended macrovesicular steatosis of hepatocytes and mononuclear cell infiltration in periportal areas at 8 months of age and, ultimately, leading to HCC development by 18 months of age [95].

Subsequent studies have revealed that both transcriptional and post-transcrip7 in cirrhotic livers of CCl_4_-treated rats and in the human HepG2 cell line is associated with the methylation of the CCGG sequence of *MAT1A* promoter [96]. In Huh7 cells, the CCGG methylation at +10 and +80 of coding region is associated with *MAT1A* down-regulation [97]. In contrast, *MAT2A* up-regulation in human HCC is associated with CCGG hypomethylation of the gene promoter [98].

Post-transcriptional mechanisms are also involved in the generation of the *MAT1A*/*MAT2A* switch, in preneoplastic and neoplastic lesions. The proteins HuR/-methyl-HuR and AUF1 regulate MAT expression during liver proliferation, differentiation, and carcinogenesis [99]. The increase of the AUF1 (AUrich RNA binding factor 1) enhances mRNA decay [100,101]. These findings were confirmed by the observation of the association of the *MAT1A*/*MAT2A* switch and low SAM levels, with CpG hypermethylation and histone H4 deacetylation of the *Mat1A* promoter, and CpG hypomethylation and histone H4 acetylation of *Mat2A* promoter in fast growing HCC of F344 rats [94]. In slowly growing HCC, induced in BN rats, low changes in *MAT1A*/*MAT2A* ratio, CpG methylation, and histone H4 acetylation of *Mat1A* promoter were found [94]. This was associated with a rise in HuR (AUrich RNA binding factor 1), which binds to AU-rich elements, inducing the stabilization of the *MAT2A* mRNA [94]. These changes are very low/absent in slowly progressing HCC of BN rats [94]. These findings indicate that the *MAT1A*/*MAT2A* switch and the decrease in SAM level may have prognostic importance for hepatocarcinogenesis. Indeed, a higher decrease of *MAT1A*/*MAT2A* gene expression and MATI/III: MAT/II activity ratios and SAM occurs in F344 HCC than in BN HCC [94]. DNA hypomethylation promotes genomic instability (GI) [102] that increases with tumor stages [103]. In human HCC, *MAT1A*:*MAT2A* ratio negatively correlates with cell growth and GI and positively correlates with cell death and DNA methylation [104]. In MAT1A KO mice, a rise in oxidative stress and GI is associated with a decrease of the Apurinic/Apyrimidinic Endonuclease 1/A [104], a protein implicated in DNA base excision repair. Indeed, SAM was found to regulate the stability of apurinic/apyrimidinic endonuclease 1 [105], involved in DNA base excision repair [106], and prevents oxidative stress and GI of de-differentiated hepatocytes in culture [105]. 

Different miRNAs may modify the *MAT1A*:*MAT2A* switch. The rise in *MAT1A* expression following, in Hep3B and HepG2 cell lines, the knockdown of miR-664, miR-485-3p, or miR-49 inhibits cell growth and induces cell death, while the subcutaneous and intra-parenchymal injection of Hep3B cells stably overexpressing the above miRNAs induces tumorigenesis in mice [107]. Also, the inhibition of *MAT2A* and *MAT2B* expression by miR-21-3p [108] or miR-203 [109] inhibits the growth and induces apoptosis of liver tumor cell lines.

## 7. Human Hepatocarcinogenesis

A valuable feature of the recent research on the alterations of signaling transduction involved in hepatocarcinogenesis is the observation that different alterations accounting for the acquisition of a susceptible phenotype to rat hepatocarcinogenesis similarly contribute to human hepatocarcinogenesis. Indeed, the results of the research on the families at risk and segregation studies on human population [110,111,112] suggest a genetic model of the predisposition to liver cancer, similar to that controlling rodents hepatocarcinogenesis, in which a major locus and multiple low-penetrance loci play a role in various circumstances [113].

Genome-wide association studies (GWAS) in different ethnic populations showed eleven single nucleotide polymorphisms (SNPs) linked to telomere length, some of which represent genetic markers with prognostic value [114,115,116,117] in DNA base excision repair. These studies examined genetic traits variants concerning oxidative stress, inflammatory and immune aspects, cell-cycle regulation, and DNA repair mechanisms that contribute to explain differences in HCC risk.

The genetic susceptibility to hepatocarcinogenesis in rodents is phenotypically evidenced by the higher propensity to progress of preneoplastic and neoplastic lesion in susceptible mice and rats. Therefore, we comparatively studied the alterations of signaling pathways in subsets of individuals with better prognosis (survival >3 years after the diagnosis), and individuals with poorer prognosis (survival <3 years).

### 7.1. Cell Cycle Deregulation

The activation of Cyclin-dependent kinases (CDKs), associated with the up-regulation of Cyclins D1, A, and E, and to pRb hyperphosphorylation [118,119], represent prognostic markers for human HCC [120]. P16^INK4A^ complexes, in the nucleus, with the kinases CDK4 and CDK6, thus inhibiting their activation by Cyclins D and consequent hyperphosphorylation of pRb (Figure 8). The CDKs form complexes with the chaperons CDC37 (cell division cycle 37) and HSP90 (heat shock protein 90) that compete with p16^INK4A^ and hind the formation of the inhibitory complexes of p16 with CDKs [121] (Figure 8). In addition, the protein CRM1 (required for Chromosome region maintenance 1; Exportin 1) complexes with the p16 effector E2F4, and transports it to the cytoplasm, thus inactivating p16 [122,123]. A comparative study of human HCC with better prognosis (HCCB) and poorer prognosis (HCCP) revealed as higher up-regulation of HSP90/CDC37 and formation of protective complexes of p16^INK4^ and nuclear export of E2F4 by CRM1 in HCCP than in HCCB [124]. Furthermore, in 60–85% of human HCC specimens, the p16^INK4A^ gene is inactivated by the GpG methylation of its promoter [125]. A recent meta-analysis showed a strong association between *GSTP1* (glutathione s-transferase, PI) and P16^INK4A^ gene promoter methylation and an increase in HBV-related HCC susceptibility [126].

### 7.2. ERK Signaling

Studies on human HCC showed that the expression of the ERK effectors increase from surrounding liver to HCC, reaching the highest values in HCCs with poor prognosis [64]. It was also found that the expression of the protein FOXM1 correlates positively with HCC proliferation and micro-vascularization, and negatively with cell death [127]. Interestingly, FOXM1 activates the SKP2–CSK1 ubiquitin ligase, and its down-regulation inhibits the ligase expression [127]. Furthermore, in variable percentages of HCCs, the promoter methylation and the downregulation of the genes encoding CDK2 inhibitors, such as *P21^WAF1^*, *P27^KIP1^*, *P57^KIP2^*, *P130*, *RASSF1A*, and *FOXO1* (Forkhead box O1), occur more frequently in HCCP then HCCB [67]. In unmethylated cases, HCCPs are characterized by a higher SKP2-CSK1 activity than HCCBs [67]. A correlation between the rate of HCC cell proliferation and micro-vascularization and promoter hypermethylation or proteasomal degradation of CDK2, parameters inversely correlated with apoptosis, was also found [67]. Also, the overexpression of the SKP2 suppressor, *HINT1* (Histidine triad nucleotide binding protein 1), and the dephosphorylation of SKP2 by CDC14B (cell division cycle 14, Saccharomyces cerevisiae homolog B) phosphatase facilitate, in HCCB, its degradation by the ubiquitin ligase (APC/C) CDH1 (Anaphase Promoting Complex/Cyclosome and its activator CDH1) [67] (Figure 2). In HCCP, CDC14B down-regulation associated with CDK2-dependent serine phosphorylation, which impedes CDH1-SKP2 interaction associated with HINT1 inactivation, hampers SKP2 degradation [67].

The role of (dual-specificity phosphatase 1) DUSP1 and its relationships with ERK1/2 in early stages of human hepatocarinogenesis are poorly known. In HCC, the expression of *DUSP1* is inversely correlated with that of *ERK1*/*2* and the proliferation rate and micro-vessel density, while DUSP1 directly correlates with tumor apoptosis [128]. Furthermore, DUSP1 expression is related to HCC prognosis, being higher in HCCB than in HCCP, in which *DUSP1* promoter hypermethylation, loss of heterozygosity at the *DUSP1* locus, and phosphorylation, followed by ubiquitination and proteasomal degradation of *DUSP1* protein, occurs [128]. These findings suggest a putative prognostic role of pERK1/2 and DUSP1 and the possibility that pERK1/2 and SKP2–CKS1 ligase cooperate [67], through DUSP1 phosphorylation and FOXM1 activation, to provide a positive feedback regulation of HCC proliferation [129] (Figure 3).

### 7.3. PI3K/AKT/mTOR Signaling

The involvement of PTEN (phosphatase and tensin homolog) down-regulation in liver diseases predisposing to human HCC, such as NASH (non-alcoholic steatohepatitis), HCV hepatitis, and HCC [130], suggests a deregulation of PI3K/AKT/mTOR signaling. PTEN down-regulation and PI3K/AKT/mTOR pathway over-activity [130] play a role in the progression of NASH and viral hepatitis to HCC. The activation of the PI3K/AKT/mTOR pathway in HCC is linked to mutation of PIK3CA [131], mutation, deletion, or downregulation of PTEN and up-regulation of IGF and EGF and of their receptors and related growth factors [132,133].

AKT activation is implicated in poor HCC prognosis [134,135,136], whereas AKT inhibition decreases in vitro growth and orthotopic implantation of HCC cells [137,138,139,140,141]. These findings indicate a crucial role of the PI3K/AKT/mTOR signaling in hepatocarcinogenesis. A phase II study showed partial remission of HCC and stable disease at 3 months of patients with HCC treated with the mTOR inhibitor Sirolimus [140].

Recent studies have shown the modifying influence of different genes in the hepatocarcinogenesis driven by the PI3K/AKT/mTOR signaling. The *SGK3* (Serum and glucocorticoid kinase 3) moderately influences the PIK3CA(E545K)/c-Met driven HCC in mice [141]. In a subset of human HCCs with poor prognosis, it has been observed that the downregulation of *Pten* synergizes with c-*Met* to promote HCC development through the mTORC2 signaling [142]. Also, in a mouse model in which mutant forms of *Pik3ca* and *Yap* are overexpressed in the liver by hydrodynamic transfection, it was demonstrated that the simultaneous activation of PI3K and Yap pathways was frequently present in human HCC [143]. Furthermore, it was recently found that Yap activates Notch signaling by upregulating Jag-1 in mouse hepatocytes and HCC cells [144].

### 7.4. The Role of MYBL2

The transfection of Huh7 liver cancer cell line with MYBL2 strongly stimulates the G1-S and G2-M phases of cell cycle, while the contrary follows MYBL2 silencing [76]. Different genes are implicated in these MYBL2 effects. They include the activation of *MDK*, which in turn activates the cell cycle and ERK1/2 and AKT signaling cascades [76].

Higher MYBL2 and *LINC* (Long intergenic non-coding RNA) are present in HCC with a mutant p53 gene than in HCC with wild-type p53 [77]. Functional experiments on hepatoma cell lines with wild-type p53 (Huh6 and HepG2) and mutant p53 (Huh7 and Hep3B) showed that MYBL2 suppression decreased proliferation, caused cell death, and induced similar levels of DNA damage in these cell lines. However, stronger growth inhibition and cell death, associated with massive DNA damage, occurred only when MYBL2 or *LIN9* (c. elegans, homolog of) silencing was associated with doxorubicin-induced DNA damage in *P53* mutant cell lines [77]. It was found that doxorubicin did not modify the MYBL2 and *LIN9* levels in the four cell lines but inactivated the LIN9-P130 complex and gradually dissociated MYBL2 from LIN9 in *P53* wild-type cells [77]. MYBL2-LIN9 was dissociated by doxorubicin in the *P53* mutant cells. The silencing of *p53* or *P21^WAF1^* eradicated the response to DNA damage, inhibited growth, and stimulated cell death in p53 wild-type cell lines. These findings indicate that, in the presence of DNA damage, the integrity of the MYBL2–LIN9 complex is essential for the survival of HCC cells with mutant *P53*.

### 7.5. The Role of Cell Cycle

In human HCC, MAT1A:MAT2A expression and MATI/III: MATII activity ratios correlate negatively with cell proliferation and genomic instability, and positively with apoptosis and DNA methylation, and MATI/III: MATII ratio strongly predicts patients’ survival length [99,104]. Further, a decrease in AUF1 protein and MAT1A-AUF1 ribonucleoprotein, and a rise in HuR protein and MAT2A-HuR ribonucleoprotein, with a consequent destabilization of MAT1A and increased stability of MAT2B, occur in human HCC [99]. Attempts to modify *MAT1A*:*MAT2A* expression ratio by forced *MAT1A* overexpression in HepG2 and HuH7 cells induced an increase of SAM level associated with a decrease of cell growth, a rise in apoptosis, and the down-regulation of *CYCLIN D1*, *E2F1*, *IKK*, *NF-kB* and of the antiapoptotic *BCL2* and *XIAP* genes, and the up-regulation of the proapoptotic *BAX* and *BAK* genes. Multivariate analyses showed that the patients’ age, the etiology, the Edmondson-Steiner grade, the MATI/III: MATII ratio, the PCNA expression, the global DNA methylation, and the genetic instability significantly contributed to patients’ survival [99]. These findings showed the post-transcriptional regulation of MAT1A and MAT2A by AUF1 and HuR in human HCC and demonstrated that a low MATI/III: MATII ratio is a prognostic marker involved in the determination of the phenotype susceptible to HCC, as well as patients’ survival.

### 7.6. The Role of DNA Repair

The DNA damage responsive machinery plays a leading role for the survival and proliferation of tumor cells. In particular, DNA-PKcs (DNA dependent protein kinase catalytic subunit) is one of the major players in the non-homologous end-joining (NHEJ) repair process. DNA-PKcs is up-regulated and associated with a poor clinical outcome in different types of tumors, including HCC [145], since it protects cancer cells against microenvironment insults and chemotherapeutic treatments. In HCCs, DNA-PKcs is positively correlated with genomic instability, microvessel density, and growth rate, while being negatively correlated with apoptosis and patient’s survival [145]. Mechanistically, it was found that DNA-PKcs transduces the effects of HSF1 (heat shock transcription factor 1) [145]. These findings suggest the DNA-PKcs could be a valuable target for the anti-neoplastic therapy.

## 8. Conclusions

The study of the genetic background of liver cancer, in rodent models, has clearly indicated a polygenic predisposition, where highly penetrant cancer-related genes and a complex network of epistatic interactions of several modifier genes contribute to determine the cancer phenotype. Population research has shown that a similar model applies to human hepatocarcinogenesis. Therefore, the detailed knowledge of the liver tumor epigenetics is fundamental for the diagnosis, prognosis, and therapy of this tumor entity. Comparative functional genetics studies identified the best-fit mouse [146] and rat [147,148] models of hepatocarcinogenesis. Through the supervised hierarchical analysis of 6,132 genes, common to rat and human liver, it was found that DNs and HCCs of BN rats, and F344 DNs clustered with human HCCB, and F344 DNs and HCCs clustered with HCCA (Figure 9). This confirms the validity of the studies on the influence of genetic predisposition to hepatocarcinogenesis on HCC prognosis in mouse and rat models. Recently, new insights on the molecular mechanisms involved in HCC pathogenesis and prognosis have been obtained by the hydrodynamic gene transfection method in mice [149]. This approach is a powerful tool to further study the HCC pathogenesis and the genetic background of the genetic predisposition to this cancer type.

## Figures and Tables

**Figure 1 cancers-11-01450-f001:**
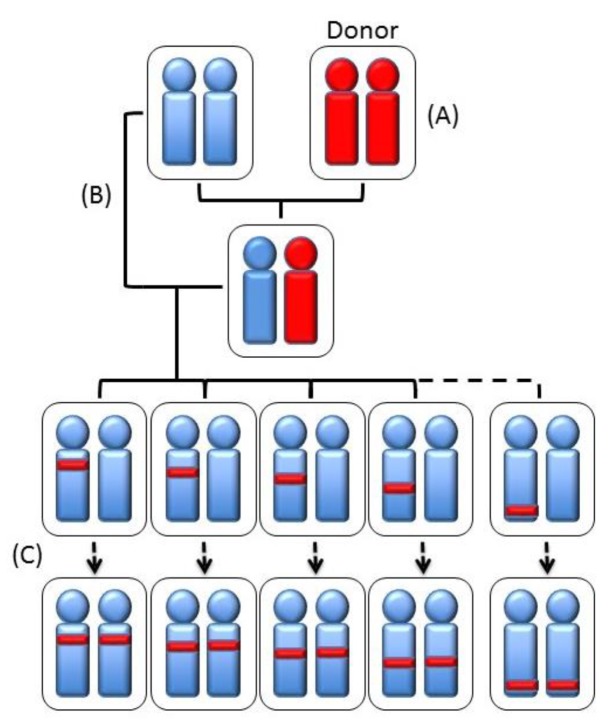
Schematic representation of the generation of recombinant congenic strains. (**A**) A donor inbred strain is selected to be crossed (**B**) with another inbred strain. The descendent is back-crossed, for 5–10 generations, with the recipient strain (**C**). Prior to each back-cross, a selection of the genotype is performed in order to pass a defined chromosomal region from the donor to the recipient strain.

**Figure 2 cancers-11-01450-f002:**
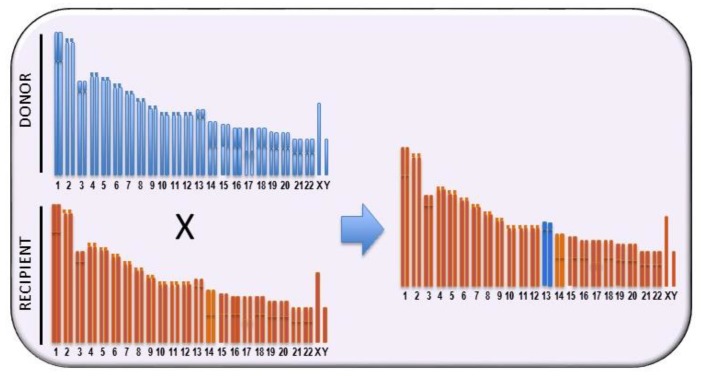
Schematic representation of the generation of an inbred consomic strain, i.e., containing a single entire chromosome from a donor strain, by backcrossing to a parental inbred strain for at least ten generations while selecting for the maintenance of a specific whole chromosome from the donor strain.

**Figure 3 cancers-11-01450-f003:**
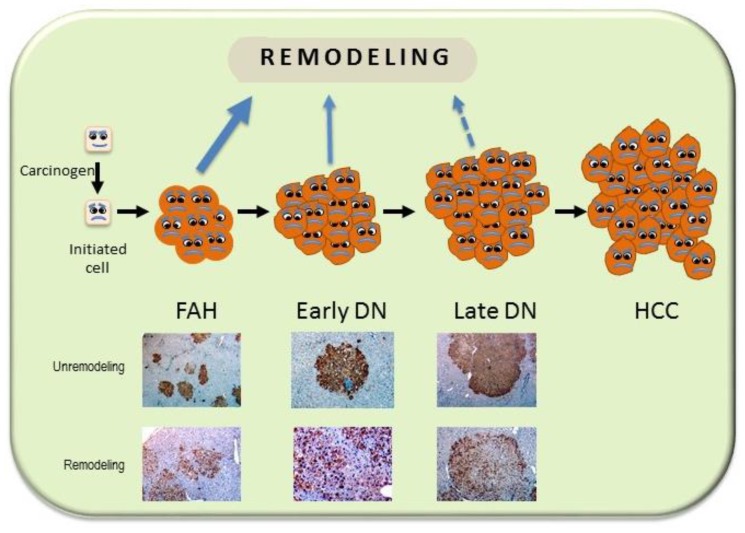
Schematic representation of multistage hepatocarcinogenesis. After hepatocarcinogenesis initiation, the initiated cells undergo a selective expansion that leads to the development of foci of altered hepatocytes (FAH), early dysplastic nodules (DN), late DN, and hepatocellular carcinomas (HCC). Severely damaged cells may undergo apoptosis (not shown), whereas weak DNA damage followed by DNA repair may consent initiated cells to re-differentiate (remodeling; reverse arrows). Remodeling progressively decreases and ceases in cancer cells.

**Figure 4 cancers-11-01450-f004:**
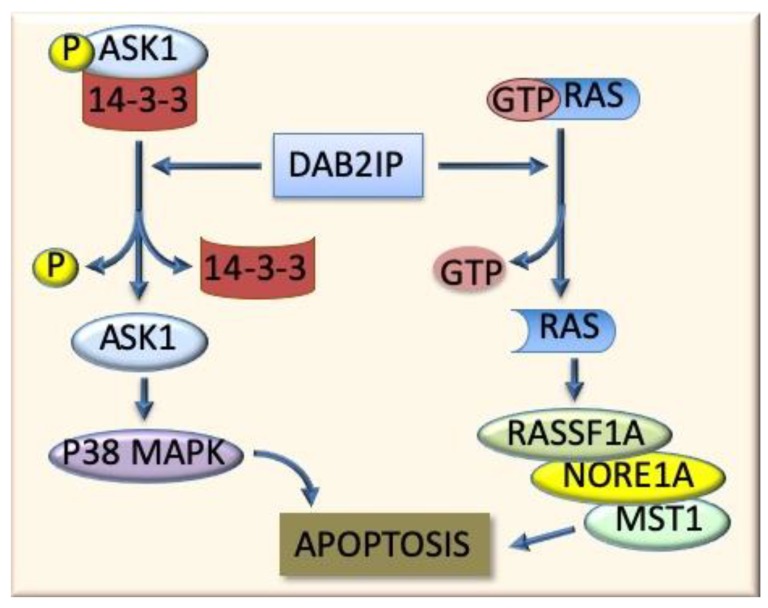
Apoptogenic role of Dab2IP protein. This protein, by impeding the formation of the complexes P-ASK1/14-3-3 and GTP/RAS, induces the activation of the apoptogenic P38 MAPK and RassF1A/Nore1A/Mst1.

**Figure 5 cancers-11-01450-f005:**
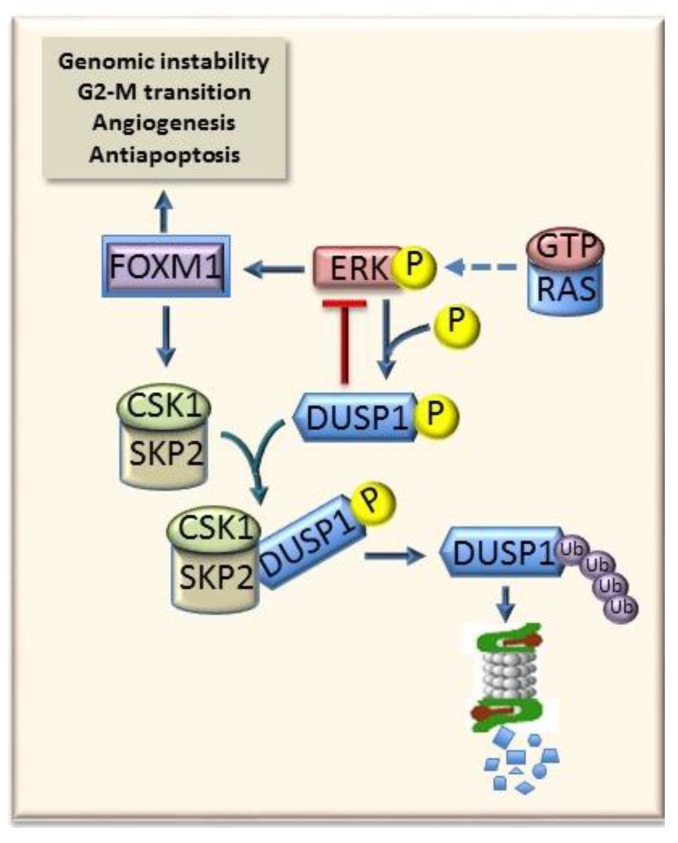
Crosstalk between RAS-ERK pathway and FOXM1. The ERK1/2 inhibitor DUSP1 may restrain the activity of RAS (RAS-GTP) and ERK1/2 signaling. The activation of FOXM1, an ERK1/2 target, and that of DUSP1, by phosphorylation of Ser296 residue allows the DUSP1 ubiquitination and proteasomal degradation. Active FOXM1 strongly influences genomic instability, cell G2-M progression, cell survival, and angiogenesis.

**Figure 6 cancers-11-01450-f006:**
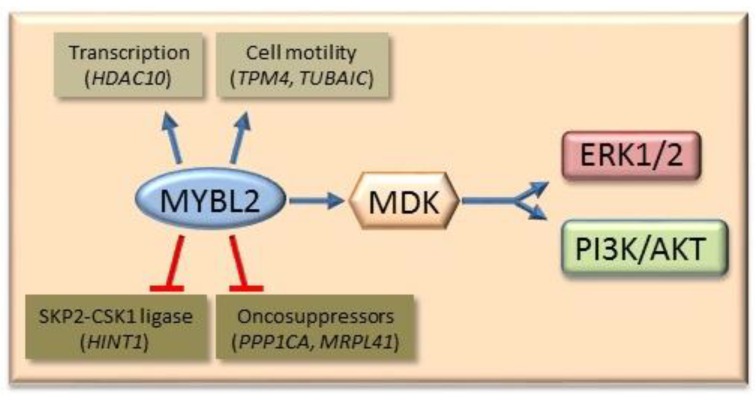
Oncogenic effect of MYBL2. MYBL2 activates different genes involved in DNA transcription, cell motility, and MDK (Midkine), an activator of PI3K/AKT and ERK1/2 pathways. Furthermore, MYBL2 inhibits of the SKP2-GSK1 ubiquitin ligase (Hint 1) and the oncosuppressors PPP1CA and MRPL41. Blue arrows indicate activation and blunt red arrows indicate inhibition.

**Figure 7 cancers-11-01450-f007:**
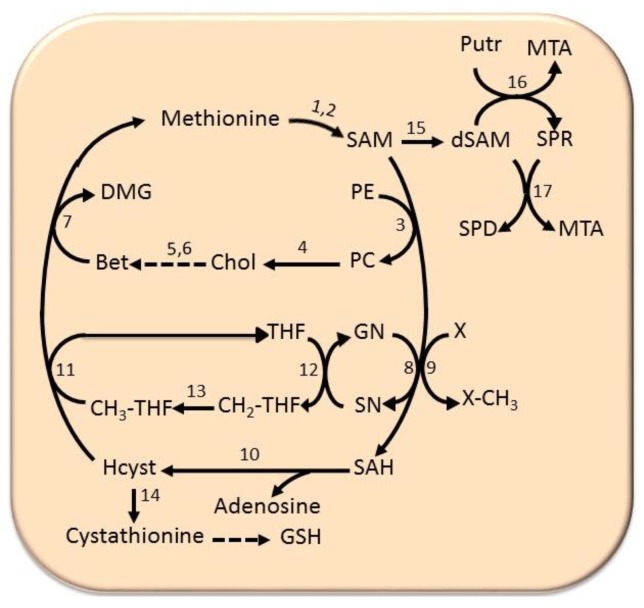
Methionine cycle. Substrates: CH_2_-THF, 5’-methenyltetrahydrofolate, CH_3_-THF, 5’-methyltetrahydrofolate, Chol, choline, DMG, dimethylglycine, dSAM, decarboxylated SAM, Bet, betaine, GN, glycine, Hcyst, homocysteine, MTA, 5’-methythioadenosine, PC, phosphatidylcholine, PE, phosphatidylethanolamine, Putr, putrescine, SAH, S-adenosylhomocysteine, SAM, S-adenosylmethionine, SN, sarcosine, SPD, spermidine, SPR, spermine, THF, tetrahydrofolate, X-CH_3_, methylated compound, X, unmethylated compound. Enzymes: 1, MATI/III; 2, MATII; 3, phospholipid N-methyltransferase; 4, various phospholipases; 5, choline oxidase; 6, betaine aldehyde dehydrogenase; 7, betaine homocysteine methyltransferase; 8, glycine N-methyltransferase; 9, various methyltransferases; 10, S-adenosylhomocysteine hydroxylase; 11, methyltetrahydrofolate reductase; 12, sarcosine dehydrogenase; 13, 5,10-methenyl-tetrahydrofolate reductase; 14, cystathionine synthetase; 15, S-adenosylmethionine decarboxylase; 16, ornithine decarboxylase; 17, spermine synthetase; 18, spermidine synthetase.

**Figure 8 cancers-11-01450-f008:**
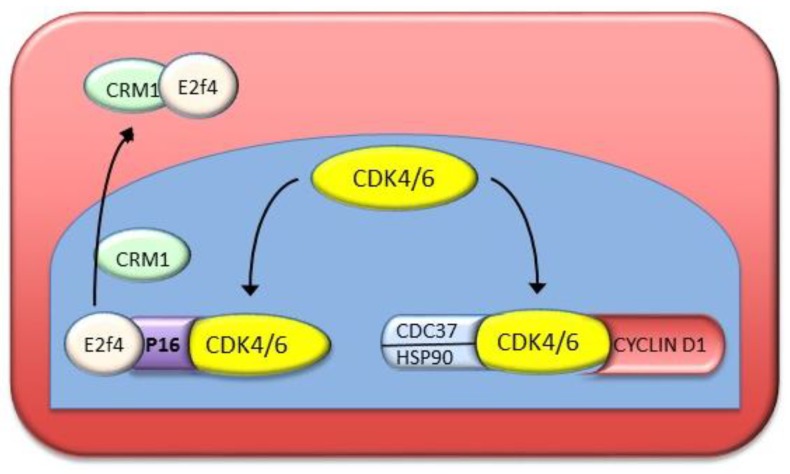
Protection by CDC37/HSP90 and CRM1 of cell cycle G1-S progression from inhibition by p16^INK4A^. In the nucleus, the formation of p16^INK4A^ complexes with CDK4 and CDK6 impedes the activity of Cyclins D. The complexes of the chaperons CDC37 and HSP90 with clyclin dependent kinases CDKs compete with p16^INK4A^, hampering the formation of the inhibitory complex. The protein CRM1 complexes the p16^INK4A^ effector E2F4, relocating it to the cytoplasm, thus inactivating p16^INK4A^.

**Figure 9 cancers-11-01450-f009:**
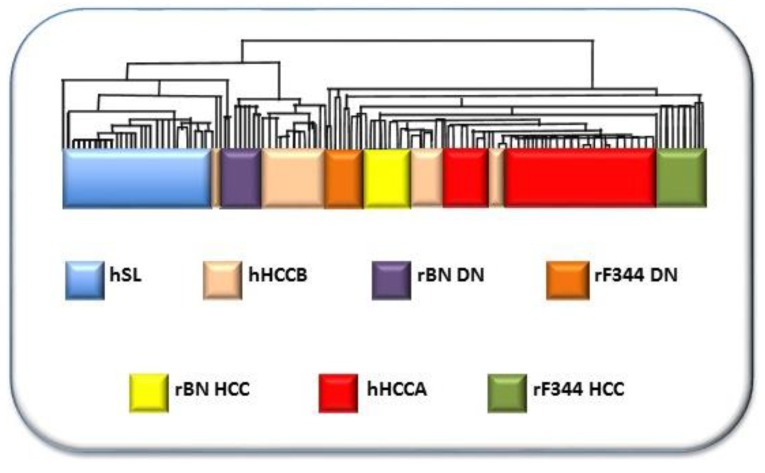
Comparative functional genomic approach by integrated unsupervised hierarchical cluster analysis of 28, 25, and 35 human surrounding non-tumorous livers, 25 HCCB and 35 HCCA, and rat liver DN and HCC. The figure is based on the hierarchical analysis of 6,132 genes, common to rats and humans, of ref. 152. Abbreviations: hSL, human surrounding liver; hHCCB, human HCC with better prognosis; rBN DN, BN rats’ dysplastic nodules; rF344 DN, F344 rats’ dysplastic nodules; rBN HCC, BN rats’ hepatocellular carcinomas; hHCCA, human hepatocellular carcinomas with poorer prognosis; rF344 HCC, F344. rats’ hepatocellular carcinomas.

**Table 1 cancers-11-01450-t001:** QTLs regulating HCC development in mice.

Strain	Chr	QTL
C3H/He x A/J	7	Hcs1
	8	Hcs2
	12	Hcs3
(C3H/He x Mus spretus) x C57BL/6J	2	Hcs4
	5	Hcs5
	19	Hcs6
RCS	1	Hcs7
Br x B6	17	Hcf1
	1	Hcf2
DBA/2J	4	Hcr1
	10	Hcr2
(C3H/He x BALB/c) F2	15	Hpcr3

**Table 2 cancers-11-01450-t002:** QTLs regulating HCC development in rats.

Strain	Chr	QTL
MHC-recombinant	20	RCC
BN × BFF1	7	Hcs1
	1	Hcs2
BFF2, CFF2, DHR	1	Hcs3/Drh1
CFF2	16	Hcs4
	1	Hcs5
	14	Hcs6
	15	Hcs7
(F344 × DRH) F2	4	Drh2a/b
BN × BFF1	10	Hcr1
	4	Hcr2
	8	Hcr3
	4	Hcr9
BFF2	8	Hcr10
CFF2	6	Hcr11
	3	Hcr12
CFF2	4	Hcr13
	18	Hcr14
BFF1	7	Lnnr1
	1	Lnnr2
	2	Lnnr3
	13	Lnnr4
	13	Lnnr5
	13	Lnnr6

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
