# Peer review of "Experimental Models to Define the Genetic Predisposition to Liver Cancer"

_cancers, 2019, doi:10.3390/cancers11101450_

Round 1

Reviewer 1 Report

This manuscript was well reviewed for experimental models to define the genetic predisposition to liver cancer.

Author Response

Reviewer n. 1:

No suggestions.

Reviewer 2 Report

Rosa M. Pascale et al summarized recent mouse and human HCC cancer research progress, focusing on the genetic mutations in the prediction of HCC. some comments:

It could be better if the manuscript section 4-7 the genes and signaling pathways classified around the hallmarks of cancer and it would be easy for the reader to understand and emphasized in the figures for each signaling and how it would contribute the HCC progression.  More adequate citations should be added. eg in section 6.1 line 312-316, the citations should be included.

Author Response

Reviewer n. 2

“It could be better if the manuscript section 4-7 the genes and signaling pathways classified around the hallmarks of cancer and it would be easy for the reader to understand and emphasized in the figures for each signaling and how it would contribute the HCC progression.  More adequate citations should be added. eg in section 6.1 line 312-316, the citations should be included.”

I thank the reviewer for the constructive comments.

The figure 4 and the legends of the figures 4, 5, 6 and 8 have been modified to better show the contribution of the related pathways to HCC development.

As it concerns the citations in the section 6.1, it must be considered that observations in c-Myc mice, i.e. the activation of TORC2 with consequent phosphorylation/activation of Akt1, but not Akt2, the fact that the loss of Akt1, but not that of Akt2, prevented c-Myc HCC formation in mice and silencing of Rictor or Akt1 in HCC cell lines with forced c-Myc overexpression inhibited p-Foxo1 expression and strongly suppressed cell growth in vitro, and the observation that the inhibition of mTORC1 in c-Myc mice, HCC progression, whereas the inhibition of  both mTORC1 and mTORC2 by MLN0128 induced apoptosis and necrosis of tumor tissue, have been found in the work of ref. 64.

Reviewer 3 Report

This review article focused on the genetic mechanisms that control the inherited predisposition to Hepatocellular carcinoma (HCC) implicating gene-gene and gene-environment interactions. Covering multiple aspect of the topic including the genes and the signaling pathways involved and the phenotypic effects of the variation of the genetic susceptibility to liver cancer. The information are well organized, the text is easy to read, and appropriated references are provide.

Specific comment:

Please correct the font and the size of the Figure 3 caption.  

Author Response

Reviewer 3

Please correct the font and the size of the Figure 3 caption.

This recommendation was communicated to the editorial office.

Editorial Office recommendation

We compared the paper with published material and found that there are some overlaps in the content. We attached the duplication report  and wonder if you may rephrase it?

We introduced different changes to the revised manuscript, evidenced in yellow. It must be taken into account,  however, that many overlaps contain scientific nomenclature that cannot be rephrased.